# BEYOND CLASSICAL DIFFUSION: BALLISTIC GRAPH NEURAL NETWORK

## ABSTRACT

This paper presents the ballistic graph neural network. Ballistic graph network tackles the weight distribution from a transportation perspective and has many different properties comparing to the traditional graph neural network pipeline. The filters propagate exponentially faster($\sigma^2 \sim T^2$) comparing to traditional graph neural network($\sigma^2 \sim T$). We use a perturbed coin operator to perturb and optimize the diffusion rate. Our results show that by selecting the diffusion speed, the network can reach a similar accuracy with fewer parameters. We also show the perturbed filters act as better representations comparing to pure ballistic ones. We provide a new perspective of training graph neural network, by adjusting the diffusion rate, the neural network's performance can be improved.

## 1 INTRODUCTION

How to collect the nodes' correlation on graphs fast and precisely? Inspired by convolutional neural networks(CNNs), graph convolutional networks(GCNs) can be applied to many graph-based structures like images, chemical molecules and learning systems. Kipf & Welling (2016) Similar to neural networks, GCNs rely on random walk diffusion based feature engineering to extract and exploit the useful features of the input data.

Recent works show random walk based methods can represent graph-structured data on the spatial vertex domain. For example, Li et al. (2017) use bidirectional random walks on the graph to capture the spatial dependency and Perozzi et al. (2014) present a scalable learning algorithm for latent representations of vertices in a network using random walks. Except for the spatial domain, many researchers focus on approximating filters using spectral graph theory method, for example,Bruna et al. (2013) construct a convolutional architecture based on the spectrum of the graph Laplacian; Defferrard et al. (2016) use high order polynomials of Laplacian matrix to learn the graphs in a NN structure model.

### 1.1 BACKGROUND AND RELATED WORK

#### 1.1.1 LAZY RANDOM WALK

Consider a undirected graph $G(V, E)$, for random walk start from vertex $v \in V(G)$, let $p_t(u)$ denotes the probability on vertex $u$ at time $t$, we have $\sum_u p_t(u) = 1$. At $time = t + 1$, the probability at vertex $v$ will be:

$$p_{t+1}(v) = \sum_{(u,v) \in E(G)} p_t(u) \cdot \frac{1}{d(u)} \qquad (1)$$

where $d(u)$ is degree on vertex $u$. The normalized walk matrix is defined as $D^{-1/2}AD^{-1/2}$. where $E(G)$ denotes the edges on $G$. Matrix notation as follows:

$$\vec{p}_{t+1} = AD^{-1}\vec{p}_t \qquad (2)$$

Consider a lazy random walk with 1/2 probability staying on current nodes. $AD^{-1}$ becomes $\frac{1}{2}(AD^{-1} + I)$ and the lazy normalized lazy walk is $(I + D^{-1/2}AD^{-1/2})/2$, where $A$ is the adjancy matrix and $D$ is the degree matrix. For regular graph, $D^{-1/2}AD^{-1/2} = AD^{-1} = D^{-1}A$.

### 1.1.2 GRAPH CONVOLUTIONAL NETWORK

Graph convolutional networks(GCN) are powerful tools for learning graph representationKipf & Welling (2016). For traditional GCN, the structure is shown in Figure 1. Akin to neural networks(NNs), promising improvements have been achieved by defining the **random walk diffusion-based filters** and using them in a multi-layer NN. However, as the depth of layers grows, over-

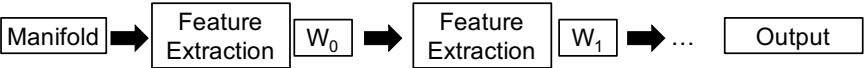

Figure 1: Structure of GCN.

smoothing appears a common issue faced by GCNs.Li et al. (2018). The over-smooth can be attributed to the stacking of random walk diffusion-based feature extraction, resulting in the similarty of node representations.

In Kipf & Welling (2016), the convolution is $\tilde{L} = I + D^{-1/2}AD^{-1/2}$. In practice, the first part can be regarded as adding self-loops to the node and then the latter part can be regarded as a walk based diffusion. For a $k$ step lazy random no biased walk, the final probability distribution of the random walk will converge to applying $\tilde{L}^k$ on the initial state. The distance from the start point of a simple random walk will converge to $C\sqrt{k}$, where $C, k$ is the constant and the number of total steps respectively. [1].

## 1.2 THE OVER-SMOOTH PROBLEM

### 1.2.1 THE LOW PASS FILTER

The random walk based method can be regarded as a low pass filter on the graph. The normalized graph Laplacian is $L_{normalized} = I - L$ where $L = D^{-1/2}AD^{-1/2}$. It is easy to prove that $L_{normalized}$ has an eigenvalue 0 with a eigenvector $\boldsymbol{d}^{1/2}$, where $\boldsymbol{d}$ is the degree vector. $L_{normalized}$ has $n$ eigenvalues: $\lambda_0 \leq \lambda_i \leq \lambda_{n-1} \leq 2$ with normalized orthonormal eigenvectors $\phi_i$, $\phi_0 = \frac{\boldsymbol{d}^{1/2}}{\|\boldsymbol{d}^{1/2}\|}$.

Fourier transformation for a signal $\pi_t$ on graph with basis $\phi_i$ is $f(\pi_t)_i = \pi_t \cdot \phi_i$, $t$ denotes the time steps. $\phi_0$ and $\lambda_0$ corresponds to the lowest frequency part and the larger $\lambda_i$ correponds to higher frequency components.

Consider the operator $L_{rw} = I + AD^{-1}$, $L_{rw}$ has $n$ eigenvectors $\phi_i^{rw} = D^{1/2}\phi_i$ and eigenvalues $\lambda_i^{rw} = 1 - \lambda_i/2$. The $\lambda_i^{rw}$ has a range between 0 and 1. The largest eigenvalue is $\lambda_0^{rw} = 1$ with the eigenvector $\phi_0^{rw}$. The normalized distribution at $t = 0$ is $\pi_0^{Nor} = D^{-1/2}\pi_0$, where $\pi_0$ is the spatial distribution[2].

The normalized distribution after $t$ steps is: $\pi_t^{Nor} = (L_{rw})^t\pi_0^{Nor}$, the Fourier transform reads:

$$f(\pi_t^{Nor}) = \sum_{k=0}^{n-1} f(\pi_0^{Nor})(\lambda_i^{rw})^t\phi_i^{rw} = f(\pi_0^{Nor})(\lambda_0^{rw})^t\phi_0^{rw} + \sum_{k=1}^{n-1} f(\pi_0^{Nor})(\lambda_i^{rw})^t\phi_i^{rw} \quad (3)$$

Since $\lambda_0^{rw} = 1$, thus the diffusion operator $(L_{rw})^t$ preserves the zero-frequency component $\phi_0^{rw}$ and suppress the high frequency part. **In this case, as the depth of GCN increase, the high-frequency information is lost, resulting in the over-smooth problem.**

### 1.2.2 THE SPATIAL DOMAIN

In this section, we analyze the long-time random walk behaviour from the spatial domain. Figure 2 (a) and (b) show the short time and long-time behaviour respectively, for short time random walk,

---

[1]there are many discussions about random walk asymptotic behaviour, for example, please see: https://www.mit.edu/~kardar/teaching/projects/chemotaxis(AndreaSchmidt)/more_random.htm, http://mathworld.wolfram.com/RandomWalk2-Dimensional.html and http://www.math.caltech.edu/~2016-17/2term/ma003/Notes/Lecture16.pdf

[2]$L_{rw}$ has similar matrices, share properties of represented linear operator $L$

neighbourhood information is captured and learned; as the time steps become larger, the probability distribution on the nodes becomes indistinguishable and increases the error in the classification task. The distribution of ballistic walk proposed in this paper is shown in Figure 2 (c), being different from the random walk method, the distribution puts more weight on the more distant nodes as time steps increase.

**Relationship between distribution and distance**    The definition of distance is:

$$Distance = \sum_{node} probablity\ on\ node \times least\ number\ of\ hoppings\ between\ start\ point\ and\ node \tag{4}$$

Figure 2 (b) and (c) show the distribution at the same number of steps start from the same point. As the ballistic walk puts more weight on the farther nodes: the indistinguishable problem is circumvented, and the distance is increased. The shape of the ballistic distribution is more oscillated, thus higher frequency information is preserved. Since the distance within the same time steps is larger, we regard the ballistic walk is a faster transportation method comparing to the random walk(Figure 2(d)). **Note the faster term does not mean the ballistic walk goes to farther nodes, both the random walk and ballistic walk reach the k-th hopping nodes in time step $k$, the ballistic walk has different weight distribution on more distant nodes.**

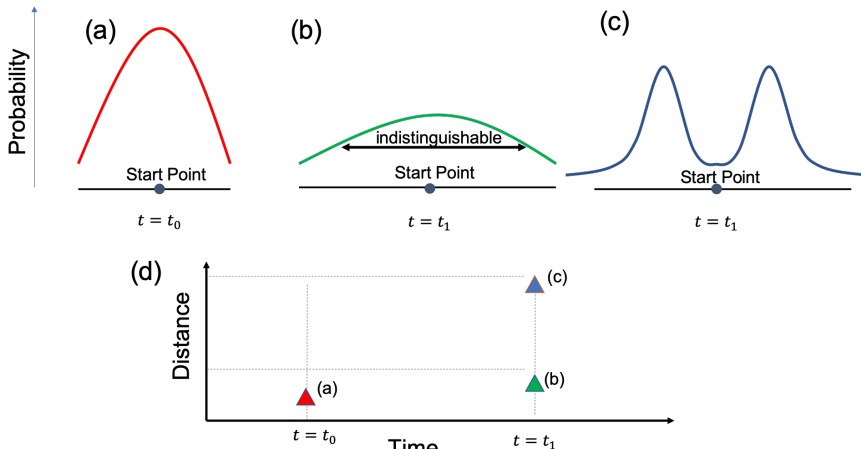

Figure 2: **(a)** Schematic weight distribution of random walk/Laplacian operator after short time steps. **(b)** Schematic weight distribution of random walk/Laplacian operator after long time steps. **(c)** Schematic weight distribution of ballistic walk after same time steps as (b). **(d)** The distance of (a),(b),(c) with their time steps respectively. (c) has a larger distance comparing to (b) at $t = t_1$, thus (c)-the ballistic walk is considered as faster transportation.

## 1.3    BALLISITC WALK IS ABLE TO COLLECT CORRLEATION BETTER

In traditional GCN, The random walk/Laplacian matrix collects the correlated information over a graph. Here we consider a two-dimensional condition, taking the start point as (0,0) and the correlated point is $(i,j)$, the distance is denoted as $d_{ij}$. As discussed, the distance walk travels is $C\sqrt{k}$, in this case, though a $d_{ij}$ step walker can reach $(i,j)$, the probability distribution on the correlated point is relatively low since the walk's average distance is $C\sqrt{k}$. In order to fully capture the correlation between the two vertices(in other words, increase the weight between (0,0) and $(i,j)$), two main methods are used:

1. take steps $d_{ij}^2$ steps, in Defferrard et al. (2016), in analogy to a 5×5 filter, the authors use Laplacian polynomial order up to 25 to achieve similar accuracy, the number of steps is far larger than the filter's size in CNNs.

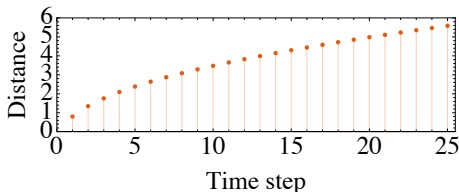

Figure 3: Classical diffusion

2. Pooling: the distance of two vertices is shortened after pooling operation. $d_{ij}$ will reduce to $\lceil d_{ij}/2 \rceil /$ after a $2 \times 2$ pooling. e.g. Henaff et al. (2015) and Bruna et al. (2013) use max pooling, Defferrard et al. (2016) use efficient pooling and Tran et al. (2018) use sort pooling.

Figure 3 shows the relation between the distance and number of steps for the random walk/Laplacian matrix. As the steps increases, the walk diffuses with the distance $\sim C\sqrt{k}$, where $k$ is the number of steps, resulting in the inefficiency in collecting information. Suggested by Hammond et al. (2011), the filter on most common graph convolutional network is:

$$g_\theta(L)x = g_\theta(U\Lambda U^T)x = Ug_\theta(\Lambda)U^T x \qquad (5)$$

$g_\theta(\Lambda) = \sum_{k=0}^{K-1} \theta_k \Lambda^k$ and $\theta \in R^K$ is a vector of polynomial coefficients, where $U \in R^{N \times N}$ comprises orthonormal eigenvectors and $\Lambda = diag(\lambda_1, ..., \lambda_n)$ is a diagonal matrix of eigenvalues, which is approximated by k-th order polynomials of $\tilde{L}$. Wu et al. (2019) Since the number of steps corresponds to the polynomial order of the Laplacian, as the polynomial order grows, the distance walker travelled changes slower and slower(the distribution becomes smooth), resulting in low efficiency and duplicated filters.

In the next section, we will introduce the ballistic walk method that, instead of walking at classical diffusion speed, the walker is able to reach an average distance $\sim Ck$ in $k$ steps walking. Different from classical diffusive transportation, this method enables us to collect correlation faster.

**Contributions** We summarize our contributions as fourfold. (1) We discuss the over-smooth of traditional GCN and propose the ballistic graph neural network. (2) We show the ballistic walk is a faster transportation comparing to classical random walk based methods. (3) We use the ballistic walk as feature extraction, and ballistic graph neural network achieves promising performance using fewer parameters comparing to random walk based feature extraction. (4) We introduce noise to the coin space during ballistic transportation. The perturbed ballistic walk transports slower and is able to collect correlation within a reasonable distance region. Thus the perturbed ballistic walk is a better representation comparing to pure ballistic filters.

## 2 BALLISTIC WALK ON GRAPH

In the following, we will focus on the regular graph to demonstrate the ballistic graph neural network, where image, video and speech data are represented.

### 2.1 INTRODUCTION TO BALLISTIC WALK

The ballistic walk algorithm consists of two parts, a walker in the position space $\mathcal{H}_{spatial}$ and a coin in the coin space $\mathcal{H}_c$. Thus the walker is described using states in Hibert space $\mathcal{H}_{spatial} \otimes \mathcal{H}_c$. Let the walker initially be at the state $|\Psi\rangle_0 = |i,j\rangle_p \otimes s_0$, where $s_0$ is normally symmetric state in $\mathcal{H}_c$. In analogy to the classical random walk, the next state of the walker can be expressed by $|\Psi\rangle_{t+1} = \widehat{\mathcal{U}} |\Psi\rangle_t$, where $\widehat{U}$ consists two operations, a flip operation $\widehat{O}_{coin}$ in the coin space and shift operation $\widehat{S}$ in the spatial space.

In this paper, we consider the ballistic walk on a regular two-dimensional graph. The coin space $\mathcal{H}_c$ consists of four states: $|\downarrow\rangle, |\uparrow\rangle, |\leftarrow\rangle, |\rightarrow\rangle$, represents move up, down, left and right for the next step. The spatial space $\mathcal{H}_{spatial}$ consists $N$ states representing the walker's position, where $N$ is

the number of nodes. The notation $|n\rangle$ denotes an orthonormal basis for $\mathcal{H}_{spatial}$ and $\langle n|$ is the Hermitian conjugate of the state. For a finite-dimensional vector space, the inner product $\langle n'|n\rangle$ is $\delta_{nn'}$ and the outer product $|n'\rangle\langle n|$ equals to a matrix in $R^{N \times N}$. The probability stay on the node $|i,j\rangle$ is $\sum_{s=\downarrow,\uparrow,\leftarrow,\rightarrow}\big\|\langle\Psi|i,j\rangle \otimes |s\rangle\big\|^2$. Pseudo-code of our method is given in Algorithm 1.

---

**Algorithm 1:** Ballistic walk on 2D regular graph

**Result:** The walker's state after K steps start from $(i,j)$

1   $p_0 = |i,j\rangle$                     // The start point

2   $s_0 = a|\downarrow\rangle + b|\uparrow\rangle + c|\leftarrow\rangle + d|\rightarrow\rangle$       // $a,b,c,d \in \mathbb{C}; \|a\|^2 + \|b\|^2 + \|c\|^2 + \|d\|^2 = 1$

3   $\widehat{S} = \sum_{i,j}|i-1,j\rangle\langle i,j| \otimes |\uparrow\rangle\langle\uparrow| + \sum_{i,j}|i+1,j\rangle\langle i,j| \otimes |\downarrow\rangle\langle\downarrow| +$
     $\sum_{i,j}|i,j+1\rangle\langle i,j| \otimes |\rightarrow\rangle\langle\rightarrow| + \sum_{i,j}|i,j-1\rangle\langle i,j| \otimes |\leftarrow\rangle\langle\leftarrow|$

4   $\widehat{O}_{coin} = \mathcal{H} \otimes \mathcal{H}$                 //$\mathcal{H}$ is usually the Hadamard matrix

5   $|\Psi_0\rangle = p_0 \otimes s_0$

6   $|\Psi_1\rangle = \widehat{S}(p_0 \otimes s_0)$

7   **for** $i = 2;\ i < K;\ i = i+1$ **do**

8     $|\Psi_i\rangle = \widehat{S}(\widehat{O}_{coin}|\Psi_{i-1}\rangle)$

9   **end**

---

## 2.2 EXPERIMENTS

**Methods** In the last section, we introduce the ballistic walk on 2D regular graph. Next, we use ballistic walk as feature extraction layer and learn graph representations. The experiment is

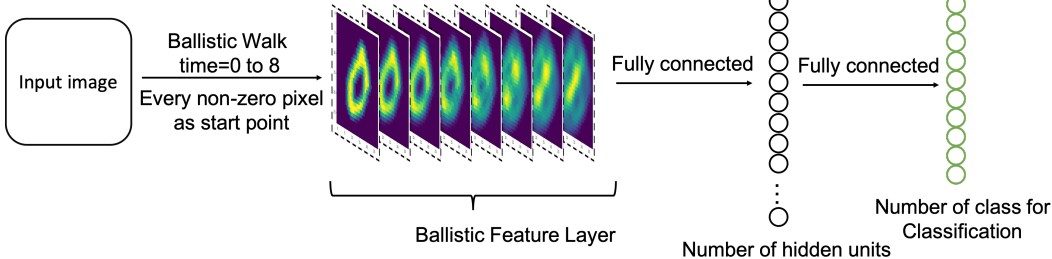

Figure 4: Schematic layout: Take every non-zero pixels as start points, the feature layer is the stack of ballistic distributions.

constructed as follows. Taking MNIST classification as an example. First, we take the non-zero pixels as the start points of the ballistic walk. The ballistic distributions at different time steps of digital **0** are shown in Figure 5. The stacked ballistic feature layer is then fully connected to a set of hidden units with relu activation. The final layer's width is the number of classes(for MNIST is 10) with softmax activation for class prediction(shown in Figure 4).

**Results** The Hadamard matrix $\mathcal{H}$ is $\frac{1}{\sqrt{2}}\begin{bmatrix} 1 & 1 \\ 1 & -1 \end{bmatrix}$ and the initial state is $\Psi_0 = 1j/2|\uparrow\rangle + 1/2|\downarrow\rangle - 1j/2|\leftarrow\rangle - 1/2|\rightarrow\rangle$. Figure 6 and 7 show the difference in diffusion between the random walk based diffusion and the ballistic diffusion on a $28 \times 28$ grid starting from the center. Comparing to the classical random walk, the ballistic walk shows cohesive behaviour and transports faster. The comparison between the speed is shown in Figure 8.

The diffusive classical walk's distances at $time = 15$ and $time = 20$ center around the same range. The ballistic walk's differences are more significant, which means collecting different information. Comparing to classical random walk, the ballistic walk has a speed of $\sim C$. This linear transportation behaviour enables the filters to collect correlation on the graph more efficiently.

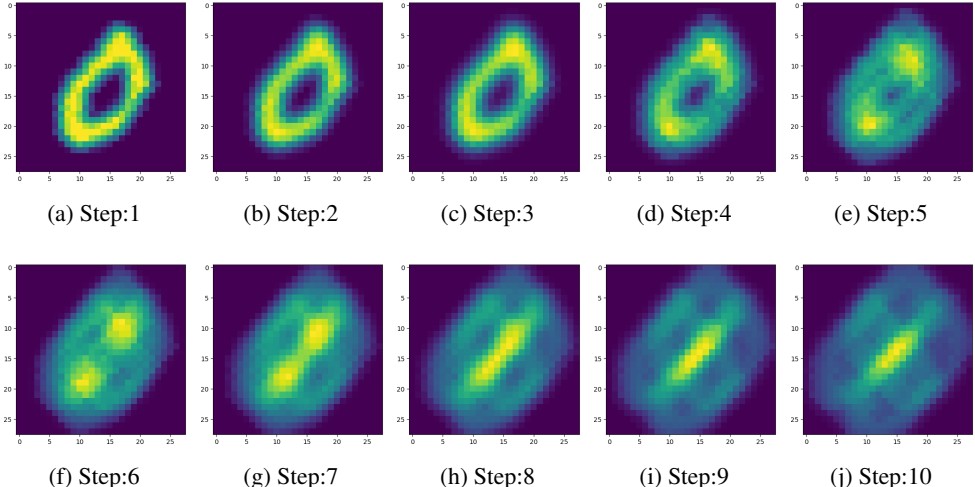

Figure 5: The ballistic distributions on digital **0** at different time steps.

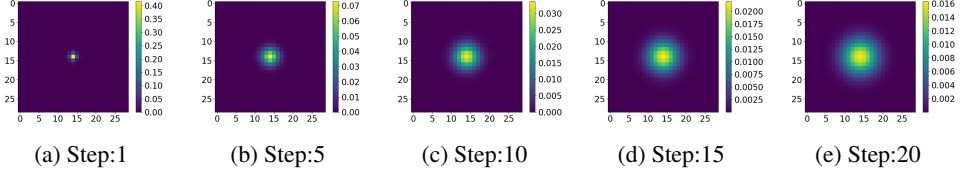

Figure 6: The classical diffusion at different steps.(starts from a point)

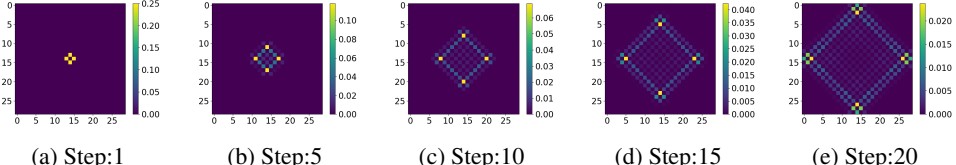

Figure 7: The ballistic diffusion at different steps.(starts from a point)

As shown in Figure 8, the distance for a diffusive walk at $time = 25$ is around taking an 8-step ballistic walk. Defferrard et al. (2016) considers 25 steps diffusive filters to approximate a $5 \times 5$ kernel with 10 feature maps(10 hidden units). For comparison, we take an 8-step-ballistic kernel with the same number of feature maps. The feature maps are then fully connected to 10/32 units and then connected to 10 units for classification. The notations are denoted as Ball10 and Ball32. Table 1 summarizes the capabilities of our model compared to other recent modeling approaches.

| Structure | Ball10 | Ball32 | GC10(Non-Param) | GC10(Spline) | GC10(Cheb) |
|---|---|---|---|---|---|
| Accuracy | **97.21**(K=8) | **97.38**(K=8) | 95.75(K=25) | 97.26(K=25) | 97.48(K=25) |
| Structure | ManiReg | DeepWalk | GCN | GAT | GLCN |
| Accuracy | 94.62 | 95.34 | 91.01 | 92.81 | 95.46 |

Table 1: Results on MNIST dataset using Ballistic filters with K = 8 compared with traditional diffusion-based graph convolutional network with K = 25 in Defferrard et al. (2016)
.

**Baselines**   We compare our approaches with the following baselines:

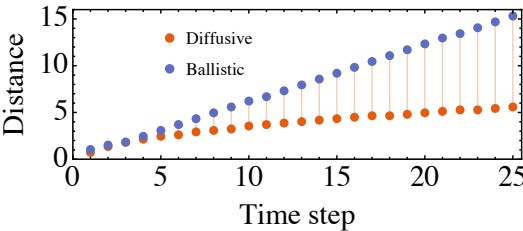

Figure 8: The diffusion behaviour of ballistic and diffusive walk.(start from a point)

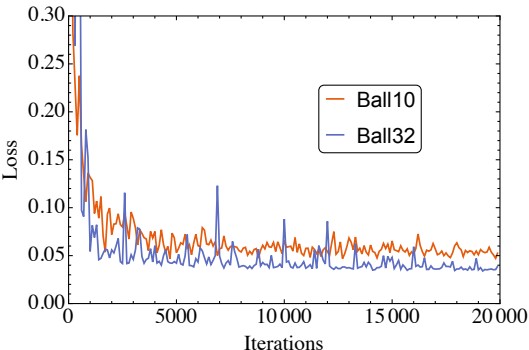

Figure 9: Results of one layer Ballistic graph network with 10 and 32 hidden units on MNIST dataset. The feature layer include 8 ballistic filters.

- **DeepWalk**: Perozzi et al. (2014) uses local information obtained from truncated random walks. For iterating over all the vertices of the graph, the authors generate a random walk $|W_{v_i} = t|$ for every node, and then use it to update representations.
- **Graph Attention Networks (GAT)**: Veličković et al. (2017) assigns different weights to different nodes in a neighbourhood. The graph attentional layer changes the weight distribution on the neighbourhood nodes.
- **Manifold Regularization (ManiReg)**: Belkin et al. (2006) brings together ideas from the theory of regularization in reproducing kernel Hilbert spaces, manifold learning and spectral methods. In the paper, their propose data-dependent geometric regularization method based on graph Laplacian.
- **Graph Convolutional Network (GCN)**: Kipf & Welling (2016) conducts the following layer-wise propagation in hidden layers using random walk based method($X^{(k+1)} = \sigma$ $(D^{-1/2}AD^{-1/2}X^{(k)}W^{(k)})$). The final perceptron layer for classication is defines as: $Z = softmax(D^{-1/2}AD^{-1/2}X^{(k)}W^{(k)})$.
- **Graph Learning-Convolutional Networks(GLCN)**: Jiang et al. (2019) contains one graph learning layer, several graph convolution layers and one final perceptron layer. The layer-wise propagation rule is: $X^{(k+1)} = \sigma(D_s^{-1/2}D^{-1/2}X^{(k)}W^{(k)})$.

## 3 REVISITING THE SPEED PROBLEM

In the last section, we introduce the ballistic walk, which transports faster than the diffusive classical walk. By selecting the ballistic filters up to $K = 8$, we reach $97\%$ and use 1/3 parameters comparing to spline method using classical diffusive filters. This suggests ballistic filters are able to collect correlation more efficiently comparing to random walk based Laplacian filters.

Figure 10 shows the transportation behaviour of different kinds of filters. There exists two phases: 'trapped to diffusive' phase and 'diffusive to ballistic' phase. The laplacian-based filters can be regarded as the up-bound filters of the trapped to diffusive phase(the orange points) and obey the $\sqrt{steps}$ law. As the steps grow, the filters are inefficient. As shown in the Figure 10, the filters are repeatedly sampling the region with distance$< 10$ as the steps grow up to 70 steps, this means the filtered information can be very similar, leading to invalid feature layer. The ballistic filters lie at the

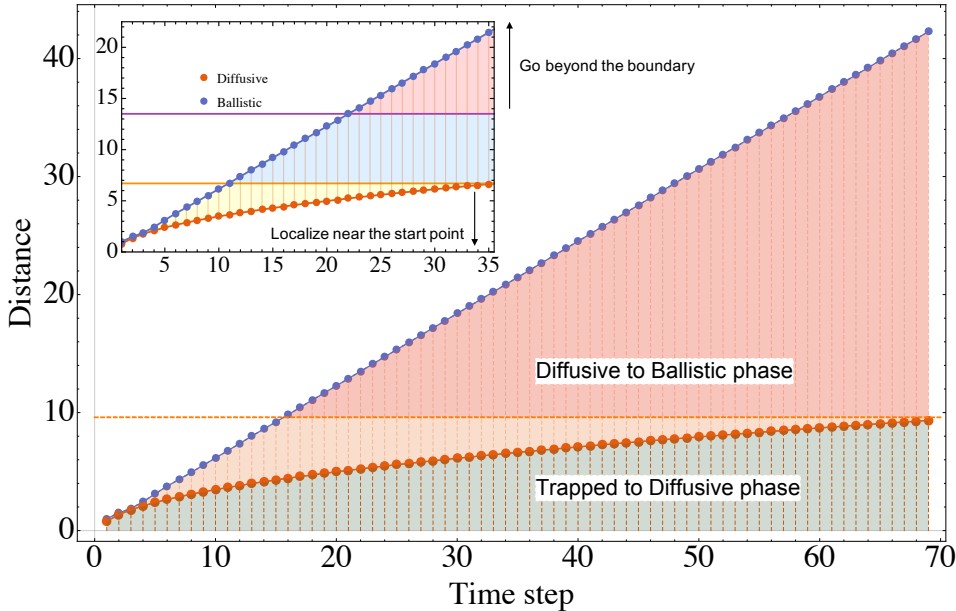

Figure 10: Comparison between diffusive and ballistic transportation. Classical diffusive walk transports slower and localizes near the start point, and ballistic walk moves beyond the boundary as steps grow. We are interested in learning the information with in the $28 \times 28$ gird.

up-bound of the 'diffusive to ballistic' phase(the blue points), the linear propagation ensures gathering the long correlation information in a relatively small number of steps. However, linear transportation also brings drawbacks:

- Sparse sampling at the mid-distance region: as shown in Figure 10 (figure in the figure), for a 35-steps walk, the points from $distance = 6$ to $distance = 13$ enables the ballistic filters better interpret the long correlation. However, the distance intervals between the ballistic filters are relatively sparse, and this can result in the missing of correlation.

- Beyond the boundary: the linear ballistic transportation makes the walker go beyond the boundary (for our case the distance is 14). With the same number of the steps, the ballistic walk travels to the boundary line(shown in Figure 11).

Is there a way to generate filters that can collect the correlation within $distance < 14$ area while circumventing cumbersome classical diffusion? In other words, we are interested in generating filters with a transportation speed between ballistic and classical diffusion. By controlling the speed of the filters, we circumvent going beyond the boundary and make all our filters localized between the regions with restricted distance(denoted as the diffusive to ballistic phase in Figure 10).

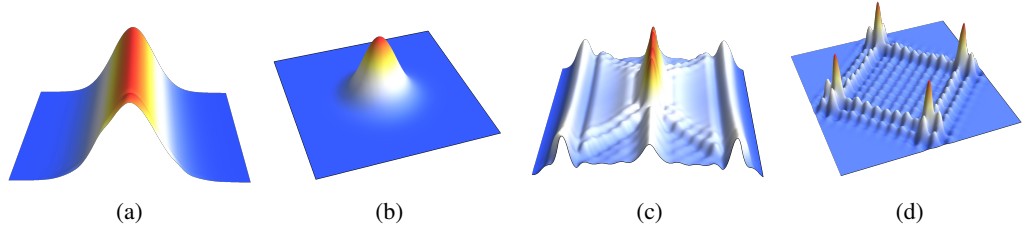

Figure 11: Comparison between ballistic and classical transportation. (a) Classical diffusion started from a line; (b) Classical diffusion started from a point; (c) Ballistic diffusion started from a line; (d) Ballistic diffusion started from a point. The ballistic filters have the exceeding boundary problem.

## 4 DE-COHERENCE

In the ballistic diffusion, we use Hadamard transformation on the coin space, The Hadamard operator (SU(2)) helps spilt the state in the coin space and finally leads to linear ballistic transportation. However, as mentioned in the last section, we are interested in generating filters lines between ballistic and classical phase so that we can circumvent the boundary and slow-transportation problem. In this section, we introduce the de-coherence scheme to perturb ballistic transportation by adding a noise term to the Hadamard operation at every step. This noisy perturbation results in the de-coherence of ballistic filters and thus slows down the transportation.

### 4.1 THE INTRODUCTION TO DECOHERENCE

We want our filters have a diffusion distance in a reasonable region($a < Distance < b$). However, the ballistic filters' distances increase with steps. The filters are not capable to dense sampling some specific regions. By selecting different randomness and steps, we can generate filters localized in a bounded area. The noisy Hadamard can be written as

$$H_r(\beta) = \begin{bmatrix} 1 & e^{i\beta} \\ e^{-i\beta} & -1 \end{bmatrix} \tag{6}$$

Table 3 shows the accuracy with different perturbed filters($\alpha = 0, 0.05, 0.10, 0.15, 0.20$). $\beta = 2 \times R \times \pi\alpha$ denotes the randomness in the coin space, $R$ is a random number between 0 and 1. The corresponding transportation speed is shown in table 2 and Figure 12. As the $\alpha$ increases to 0.20, the speed drops to

| Randomness | $\alpha = 0$ | $\alpha = 0.01$ | $\alpha = 0.05$ | $\alpha = 0.1$ |
|---|---|---|---|---|
| Speed | 0.612 | 0.608 | 0.586 | 0.516 |
| Randomness | $\alpha = 0.15$ | $\alpha = 0.20$ | $\alpha = 0.25$ | $\alpha = 0.30$ |
| Speed | 0.414 | 0.323 | 0.268 | 0.240 |

Table 2: Diffusion rate with different randomness

0.323. $\alpha$ is a controller of the diffusion speed, as $\alpha$ becomes larger, the ballistic tranportation will finally evolve to the classical diffusive couterpart.

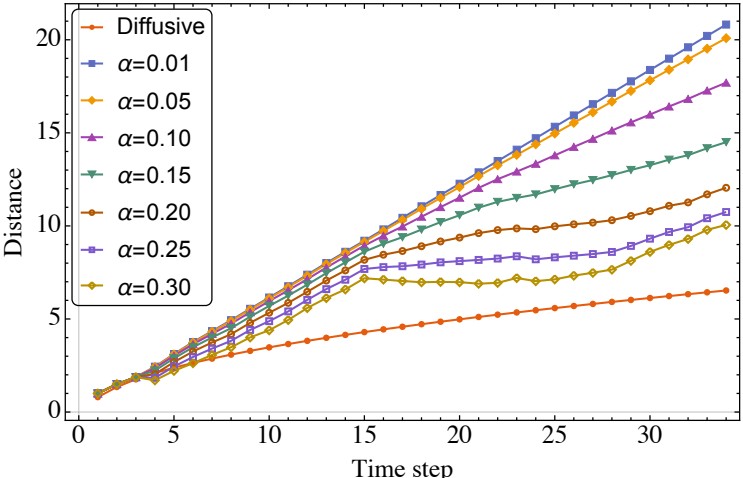

Figure 12: Comparison between diffusive and ballistic transportation with different random Hadamard operator.

### 4.2 SUMMARY OF THE SPEED WITH RANDOMNESS

After taking randomized operations, the accuracy can be improved. In other words, by using filters from the perturbed ballistic walk, we are now able to dense sample the 'meaningful regions' and avoid the shallow sampling and slow transportation problem by selecting the step and the randomness of the ballistic walk. The 'meaningful regions' are denoted as blue and yellow in Figure 10 (figure in

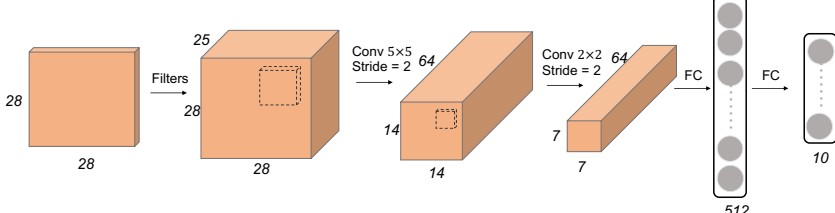

Figure 13: The model architecture with 25 filters as feature maps

the figure). In our model, we fix the first eight filters as the pure ballistic filters without perturbation. We then select different filters from perturbed filters. The model architecture is shown in Figure 13. The input signals are first passed to 25 different feature maps using the selected filters. We then apply the convolutional operation and average pooling on the feature maps. After a fully connected layer with 512 hidden units, the network is connected to 10 units for the classification task. We select our filters up to the distance<14 regions($\sim$ 25 steps for ballistic duffusion), and this ensures the filters gather the correlation information within reasonable regions. For pure ballistic filters,

| Accuracy \ Filter Index | Ballistic($\alpha=0$) | $\alpha=0.1$ | $\alpha=0.15$ | $\alpha=0.05$ | $\alpha=0.20$ |
|---|---|---|---|---|---|
| **99.11±0.13** | 1-25 | 0 | 0 | 0 | 0 |
| **99.14±0.17** | 1-8 | 9,10 | 24,25 | 10-23 | 0 |
| **99.32±0.09** | 1-8 | 3-10 | 4-10 | 8,9 | 0 |
| **99.35±0.07** | 1-8 | 1-8 | 1-8 | 9 | 0 |
| **99.23±0.13** | 1-8 | 2,4,6,8,10,12 | 3,6,9,12,15,18 | 2,4,6,8,10 | 0 |
| **99.32±0.06** | 1-8 | 3,6,9,12 | 3,6,9,12 | 3,6,9,12 | 3,6,9,12,15 |
| **99.39±0.09** | 1-8 | 3,6,9,12 | 3,6,9,12 | 9,12,15,18 | 12,15,18,21,24 |

Table 3: Accuracy with different randomized Hadamard operations. We use 25 filters with different steps and randomness for each case.

the classification accuracy is around 99.11%, when we keep the first eight ballistic filters and use different filters with different randomness, the accuracy increases to 99.39%. Our results show that the classification accuracy can be improved using a mixture of perturbed filters.

### 4.3 COIN OPERATOR

The ballistic walk filters can also be generalized to different coin operators. Except using Hadamard and noisy Hadamard coin operator, we can also use a discrete Fourier operator(DFO) or Grover operator, the discrete Fourier operator is written:

$$DFO = \frac{1}{d^{1/2}} \begin{bmatrix} 1 & 1 & 1 & ... & 1 \\ 1 & \omega & \omega^2 & ... & \omega^{d-1} \\ 1 & ... & ... & ... & ... \\ 1 & \omega^{d-1} & \omega^{2(d-1)} & ... & \omega^{(d-1)(d-1)} \end{bmatrix} = \frac{1}{2} \begin{bmatrix} 1 & 1 & 1 & 1 \\ 1 & \omega & \omega^2 & \omega^3 \\ 1 & \omega^2 & \omega^4 & \omega^6 \\ 1 & \omega^3 & \omega^6 & \omega^9 \end{bmatrix} \quad (7)$$

where $\omega = e^{2\pi i/d}$ is the $d$th root of unity, $d$ is the degree of regular graph, and the Grover operator is:

$$G = \begin{bmatrix} a & b & b & ... & b \\ b & a & b & ... & b \\ ... & ... & ... & ... & ... \\ b & b & b & ... & a \end{bmatrix} = \frac{1}{2} \begin{bmatrix} -1 & 1 & 1 & 1 \\ 1 & -1 & 1 & 1 \\ 1 & 1 & -1 & 1 \\ 1 & 1 & 1 & -1 \end{bmatrix} \quad (8)$$

where $a = (2/d) - 1$ and $b = 2/d$. The results are shown in Table 4. Note that we usually select unitary operation in the coin space to keep the probability as a constant. However, not every unitary operator results in ballistic transportation, the Grover operator will localize near the start point as the steps grow(shown in Figure 14), however, they all have a speed-up effect comparing to the classical diffusive filters. The DFO and Grover operator have a transportation speed between the ballistic

| Structure | DFO10 | DFO32 | Grover10 | Grover32 |
|-----------|-------|-------|----------|----------|
| Accuracy  | 97.32 | 97.58 | 97.26    | 97.39    |

Table 4: Performance using Grover and DFO filters using NN structure in Figure 4.

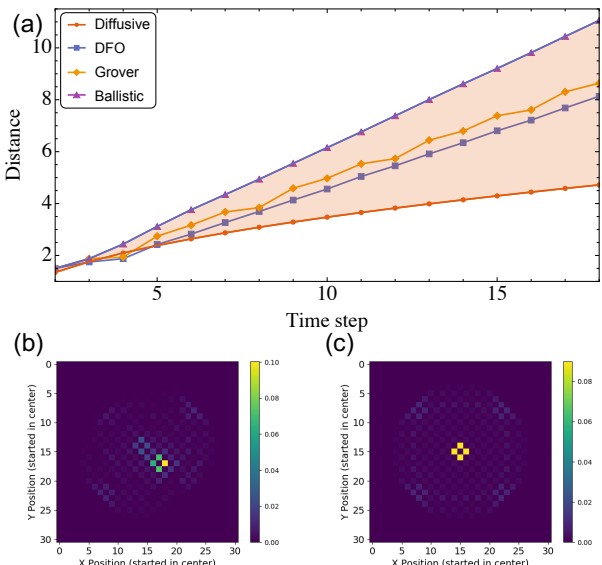

Figure 14: Comparison between DFO and Grover operator at 15 steps. (a) The transportation speed. (b) The walk's distribution after 15 steps using DFO. (c) The walk's distribution after 15 steps using Grover operator.

filters and diffusive filters, thus can be regarded as special forms of randomized filters. The general Hadamard coin is balanced. DFO retains the balanced properties on a general graph. Every state in coin space is obtained with equal probability. The Grover operator helps retain the symmetry of the signal and is permutation symmetric. The Grover operator is not a balanced coin because the probability(weight in our case) does not change its propagating directions($p = (1 - 2/d)^2$).

## 5 CONCLUSION AND FUTURE WORK

In this paper, we introduced a generalization of graph neural network: ballistic graph neural network. We started from the speed problem of the traditional diffusive kernel and tackle this problem from the perspective of transportation behaviour. We showed the linear transportation behaviour of ballistic filters and introduced the de-coherence scheme to adjust the filters' speed. Compared with diffusive filters, the ballistic filters achieve similar accuracy using fewer of the parameters. Besides, we showed that the efficiency of the ballistic filters could be improved by controlling transportation behaviour. Compared to the random walk method, we used two operators: the coin operator and shift operator to control the walker, and thus controlled the information the walker gathers. Our pipeline provides a new perspective for efficient extracting the graphical information using diffusion-related models.

Future work can investigate these two directions:

**The Network Structure.** In this paper, we use simplified architecture to demonstrate the concept of the ballistic walk, the layers are limited to 5 layers, and we use traditional average pooling. More layers can be added to improve particular accuracy, and more sophisticated pooling methods can be introduced Defferrard et al. (2016). Other techniques like dropout can also be employed to improve accuracy.

**The Ballistic Filter.** De-coherence can also be introduced into the shift operator. In other words, we can use perturbed shifted operator, and thus we introduce randomness in the spatial domain. We

can also try different unitary operators in the coin space or change the initial state of the walker. The extension to general graphs can be generalized by adding self-loops to the nodes and thus make the graph regular.

## 6    DISCUSSION: BALLISTIC FILTER IN ONE DIMENSIONAL CONDITION

The ballistic filters are inspired by two-dimensional quantum walk. The quantum coherence effect guarantees fast ballistic transportation. The different states in the coin space can be regarded as the independent state from spatial behaviour, for example, the spin of fermions or the polarization of light. More information about the quantum walk can be found at Childs et al. (2003).

Why introducing ballistic filters results in better performance? We here offer a conjecture from the perspective of signal processing using one-dimensional condition.

The classical diffusion in the one-dimensional case has the shape of:

$$g(x) = \frac{a}{\pi} e^{-ax^2} \tag{9}$$

and the frequency part can be written as:

$$\hat{g}(f) = e^{-\frac{\pi^2 f^2}{a}} \tag{10}$$

The $g(x)$ can be regarded as a gaussian low pass filter. For a gaussian high-pass filter, the spatial distribution is: Makandar & Halalli (2015)

$$hg(x) = C(1 - e^{\frac{-(\frac{H}{2}-x)^2}{A^2}}) \tag{11}$$

The long time probability distribution of ballistic walk is: Luo & Xue (2015)

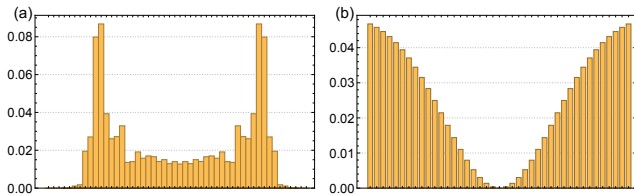

Figure 15: Different distributions. (a) cumulative distribution of 24th and 25step of ballistic diffusion. (b) Gaussian high pass filter.

$$P(x) = P_0 + ae^{-d\frac{(x-b)^{1.5}}{N^{0.5}}} \tag{12}$$

Figure 15 shows the distribution of gaussian high pass filter and the cumulative distribution of 24th and 25step of ballistic diffusion. These two distributions have a similar shape while the ballistic distribution has steeper edges resulted from fast transportation.

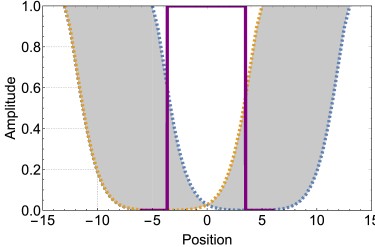

Figure 16: Ballistic diffusion with pulse signal.

The ballistic filters' capability to collect the long-time probability means it can act as a high-pass filter with different sizes. The size of the filters depends on the walking steps. Figure 16 shows the ballistic

diffusion with a pulse signal from $t = -\tau$ to $t = \tau$. The orange dashed line is an approximated shape of ballistic transportation of the leftmost signal($t = -\tau$), and the blue dashed line corresponds to $t = \tau$. The width of the approximated shape is related to the walking steps. For random walk based diffusive transportation after certain steps of diffusion, the region from $t = -\tau$ to $t = \tau$ have a gaussian shape since it is sum of gaussian distribution with centers range from $t = -\tau$ to $t = -\tau$. The classical diffusion acts like a blur filter(low pass filter). For ballistic diffusion, the shape of the pulse signal from $t = -\tau$ to $t = \tau$ evolves to a 'valley' shape and thus, the ballistic diffusion is similar to a high pass filter.

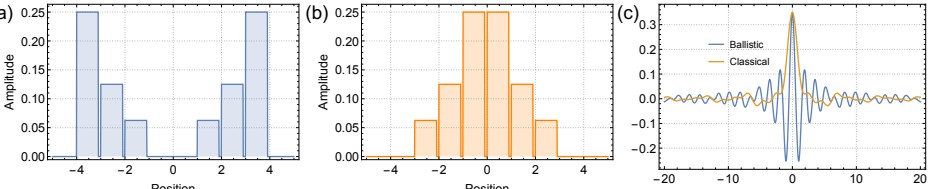

Figure 17: Schematic diagram of different filters and their Fourier transformation.

Figure 17 (a) and (b) shows approximate shape of the one dimensional ballistic and classical filtering result with a pulse signal from $t = -3$ to $t = 3$, respectively. Figure 17(c) shows the Fourier transformations. With faster transportation, the filters are capable of collecting more high frequency compared to localized diffusive filters.

**Acknowledgement**   We thank Nvidia for donating NVIDIA DGX-1 used for this research

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

# Responses to Reviewers

## on Paper *Beyond Classical Diffusion: Ballistic Graph Neural Network*

**Referee #3 and #2:**
**More experiments to demonstrate the power of ballistic comparing to classical ones.**
**Classification on different DATASETs** (accuracy %)

| Name of dataset | Classical Diffusion based ([1]) | Classical Diffusion based ([2]) | Ballistic (this paper) |
| --- | --- | --- | --- |
| CIFAR-10 | 93.65 | 93.49 | ***94.73*** |
| CIFAR-100 | 60.30 | 59.51 | ***61.24*** |
| STL-10 | 55.28 | 58.87 | ***63.32*** |

**[1]** Thomas N Kipf and Max Welling. Semi-supervised classification with graph convolutional networks. *arXiv preprint arXiv:1609.02907*, 2016.

**[2]** Michaël Defferrard, Xavier Bresson, and Pierre Vandergheynst. Convolutional neural networks on graphs with fast localized spectral filtering. In *Advances in neural information processing systems*, pp. 3844–3852, 2016.

**Referee #1:**

**If the contribution is only a new kind of random walk on a graph, is ICLR the good targeted venue ?**

**Response:** Answer: Yes, random walk on graph is a very important problem. In GCN, random walk is the way you collect the information. The traditional pipeline is show attached.

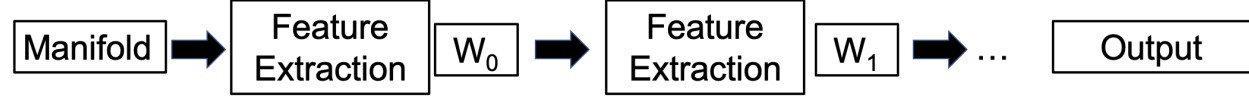

*The Feature Extraction lies as the **cornerstone** of the GCN problem:*
*For almost all previous papers, the methods used are based on a random walk (Laplacian matrix), resulting in the over-smooth problem as* **Referee #3** *mentioned. The method we propose in this paper is inspired by quantum diffusion and is completely different from the previous one. Our method also achieves better performance.*

**Referee #1 and #2:**

**- The proposed algorithm is not clear.**

Response: In light of the reviewer's recommendation, we now present the algorithm in one-dimensional condition for better understanding and ballistic concept.

- **The ballistic concept is not introduced at all in section 4. Referee #1**

    Response: The **ballistic** concept is taken from Condensed Matter physics.(please see the introduction in this link http://asdn.net/asdn/electronics/transport.php).
    For Electron Transport in Semiconductors. There are three types of transportation behaviour: Diffusive, quasi- Ballistic and Ballistic. Please see the attached figure. The 'Ballistic' means transportation is exponentially accelerated.

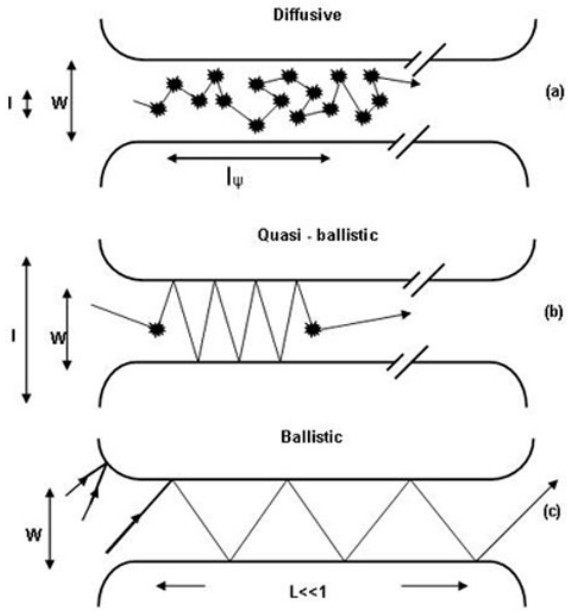

- **The proposed algorithm.**
    Response: Here we demonstrate the algorithm in one-dimensional case, this algorithm is also called quantum walk, for one-dimensional case, please see here: Section

    Discrete Time Quantum Walks https://en.wikipedia.org/wiki/Quantum_walk
    There are two operators, a "coin flip" operator and a conditional shift operator, which are applied repeatedly.
    1. First, the distribution on one-dimensional line (9 nodes: $x = -4 \ to \ 4$) is:

$$\{0,0,0,0,1,0,0,0,0\}$$

The state on the middle node (x=0) is $\frac{1}{\sqrt{2}}$ ($|left\rangle + i\,|right\rangle$). The left state means go left and the right state means go right, the state is in the coin Hilbert space. $|left\rangle$ corresponds to $\binom{1}{0}$ and $|left\rangle$ corresponds to $\binom{0}{1}$ .

2. ***(Coin space operation)*** Then we apply the coin operator, in one-dimensional case the coin operator is the Hadamard Gate. $H = \frac{1}{\sqrt{2}} \begin{pmatrix} 1 & 1 \\ 1 & -1 \end{pmatrix}$, apply coin operator on the state

$H \frac{1}{\sqrt{2}}$ ($|left\rangle + i\,|right\rangle$) $= \frac{1}{\sqrt{2}} H \binom{1}{i} = \frac{1}{2} \binom{1+i}{1-i} = \frac{1}{2} ((1+i)|left\rangle + \frac{1}{2} (1 - i)|right\rangle$

***(Spatial space operation)*** Now the line is $\{0,0,0,0,1,0,0,0,0\}$ with a state $\frac{1}{2} (1 + i)|left\rangle + \frac{1}{2} (1 - i)|right\rangle$ on $x = 0$. For every node, move the $|left\rangle$ state amplitude to left node and $|right\rangle$ to right node.

Now the line is $\{0,0,0,\frac{1}{2} (1 + i), 0, \frac{1}{2} (1 - i), 0,0,0\}$, with pure state $|left\rangle$ on $x = -1$ and pure state $|right\rangle$ on $x = 1$.

The probability is the norm (multiply the conjugate complex number) of the state amplitude: the distribution on a line becomes:

$$\{0,0,0,\tfrac{1}{2}\ 0,\tfrac{1}{2},0,0,0\}$$

with pure state $|left\rangle$ on $x = -1$ and pure state $|right\rangle$ on $x = 1$.

3. ***Coin space operation:*** the amplitude distribution on a line last moment: $\{0,0,0,\frac{1}{2} (1 + i), 0, \frac{1}{2} (1 - i), 0,0,0\}$

Using coin operator : $H \frac{(1+i)}{2} \binom{1}{0} = \frac{1+i}{2\sqrt{2}}\binom{1}{1}$ on $x = -1$ and $H \frac{(1-i)}{2} \binom{0}{1} = \frac{1-i}{2\sqrt{2}}\binom{1}{-1}$. For $x = -1$, $\frac{1+i}{2\sqrt{2}}\binom{1}{1} = \frac{1+i}{2\sqrt{2}}|left\rangle + \frac{1+i}{2\sqrt{2}}|right\rangle$, moves the $|left\rangle$ state amplitude to left node and $|right\rangle$ to right node. Same as For $x = 1$.

***(Spatial space operation)*** the amplitude on the line becomes

$$\{0,0,\tfrac{1+i}{2\sqrt{2}}|left\rangle,0,\tfrac{1+i}{2\sqrt{2}}|left\rangle + \tfrac{1-i}{2\sqrt{2}}|right\rangle,0,\tfrac{1-i}{2\sqrt{2}}|right\rangle,0,0\}=$$
$$\{0,0,\tfrac{1+i}{2\sqrt{2}},0,\tfrac{1}{\sqrt{2}},0,\tfrac{1-i}{2\sqrt{2}},0,0\}$$

With $|left\rangle$ at $x = -2, \frac{1}{2}((1 + i)|left\rangle + (1 - i)|right\rangle)$ at $x = 0$ and $|right\rangle$ at $x = 2$.

**The probability is the norm (multiply the conjugate complex number)**
$$\{0,0,\tfrac{1}{4},0,\tfrac{1}{2},0,\tfrac{1}{4},0,0\}$$
Repeat the two operation again and again.

Here we plot ***Probability*** (not amplitude) evolution on a line with 21 nodes.

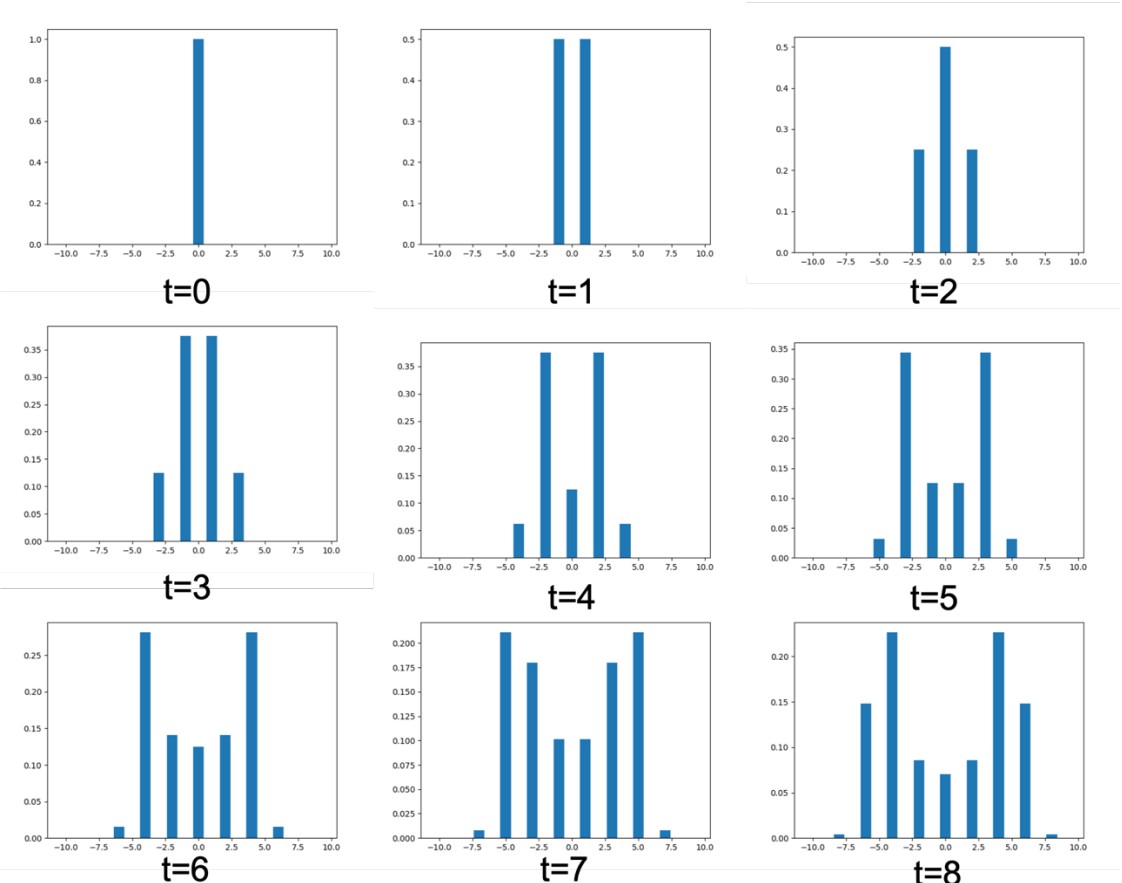

**Referee #3 the over smooth problem:**

The smooth problem means after apply traditional Laplacian-based matrix on a manifold; the distribution will become more and more **'flat'** on the spatial domain, as shown in the figure below:

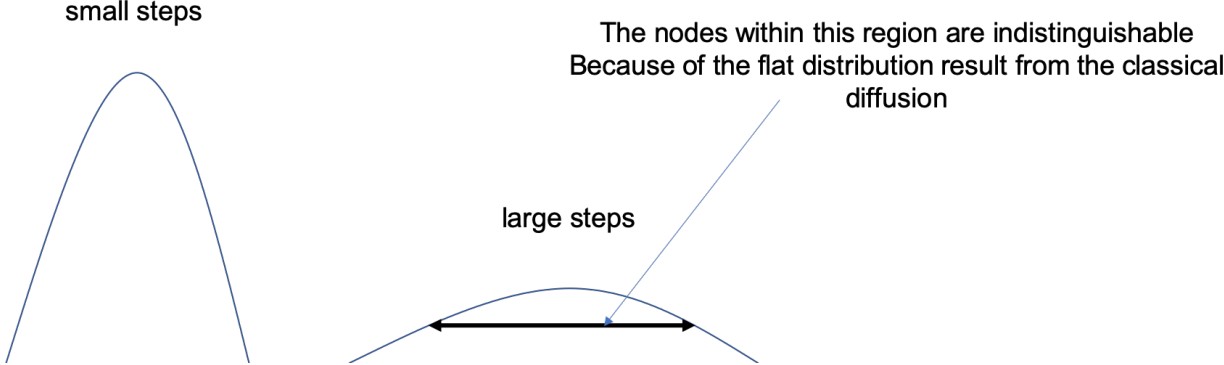

But the ballistic diffusion avoids this problem, see the attached figure:

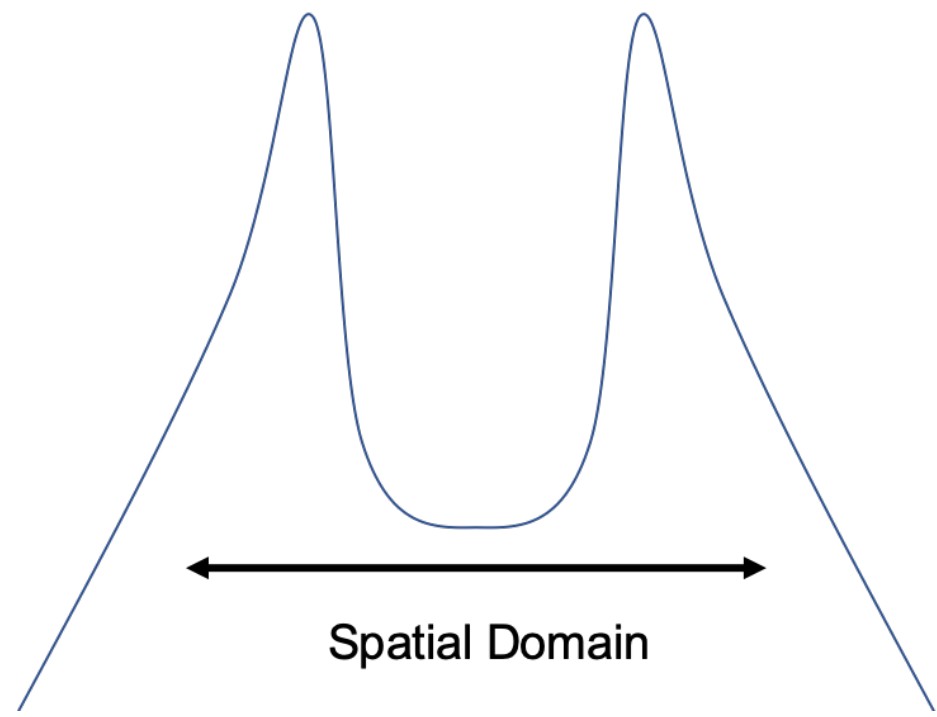

# Ballistic diffusion on MNIST dataset:
(step 0 to 10)

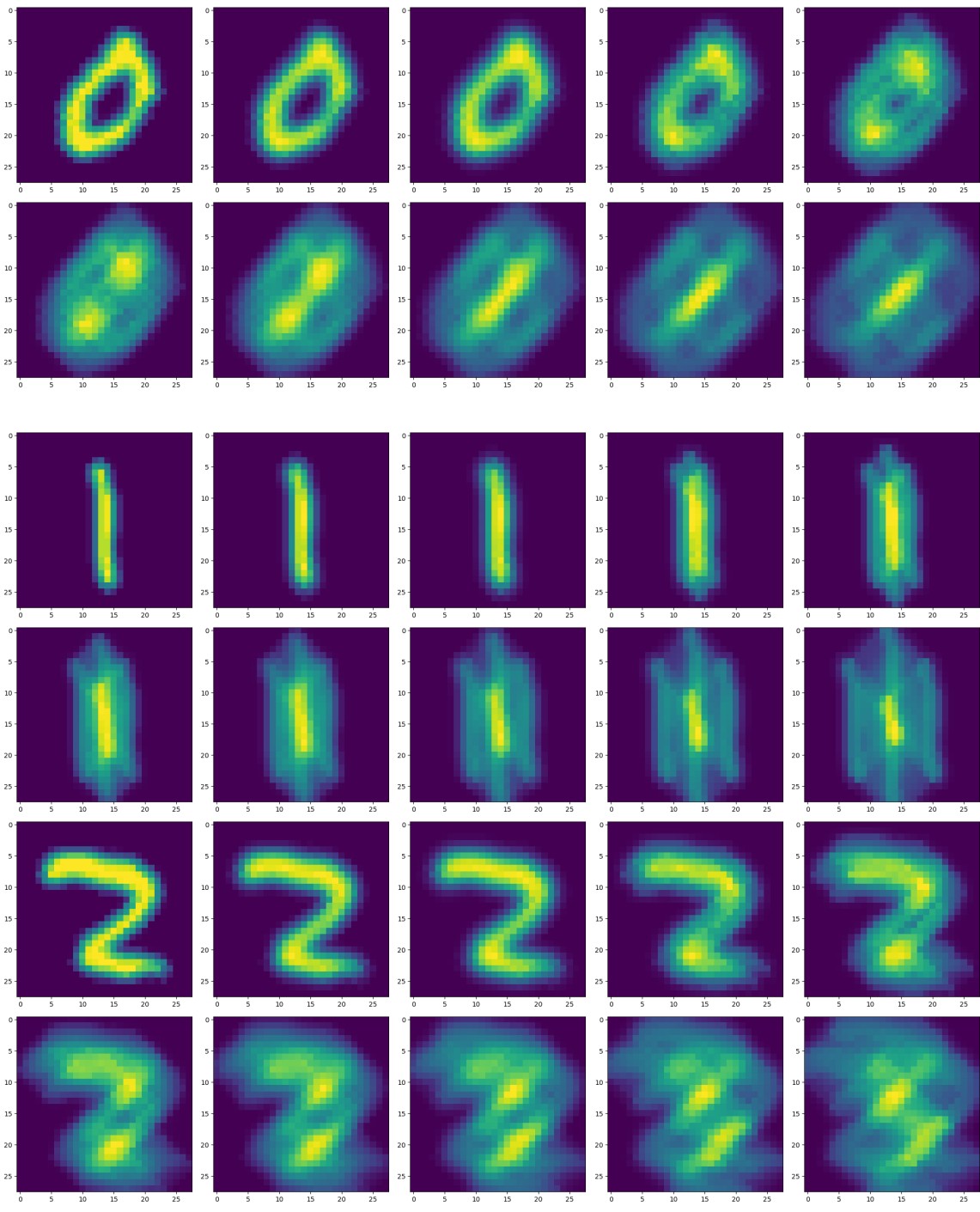

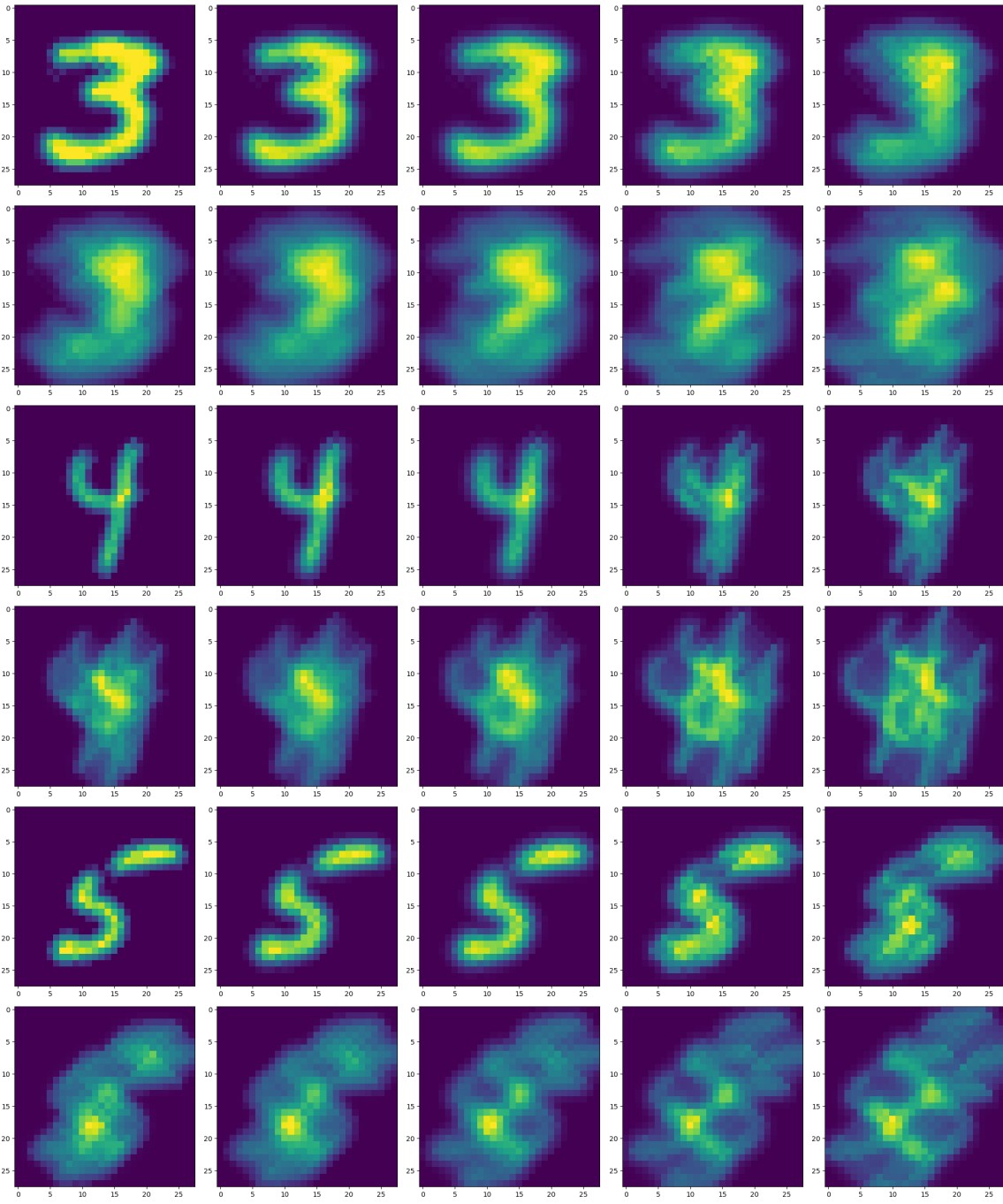

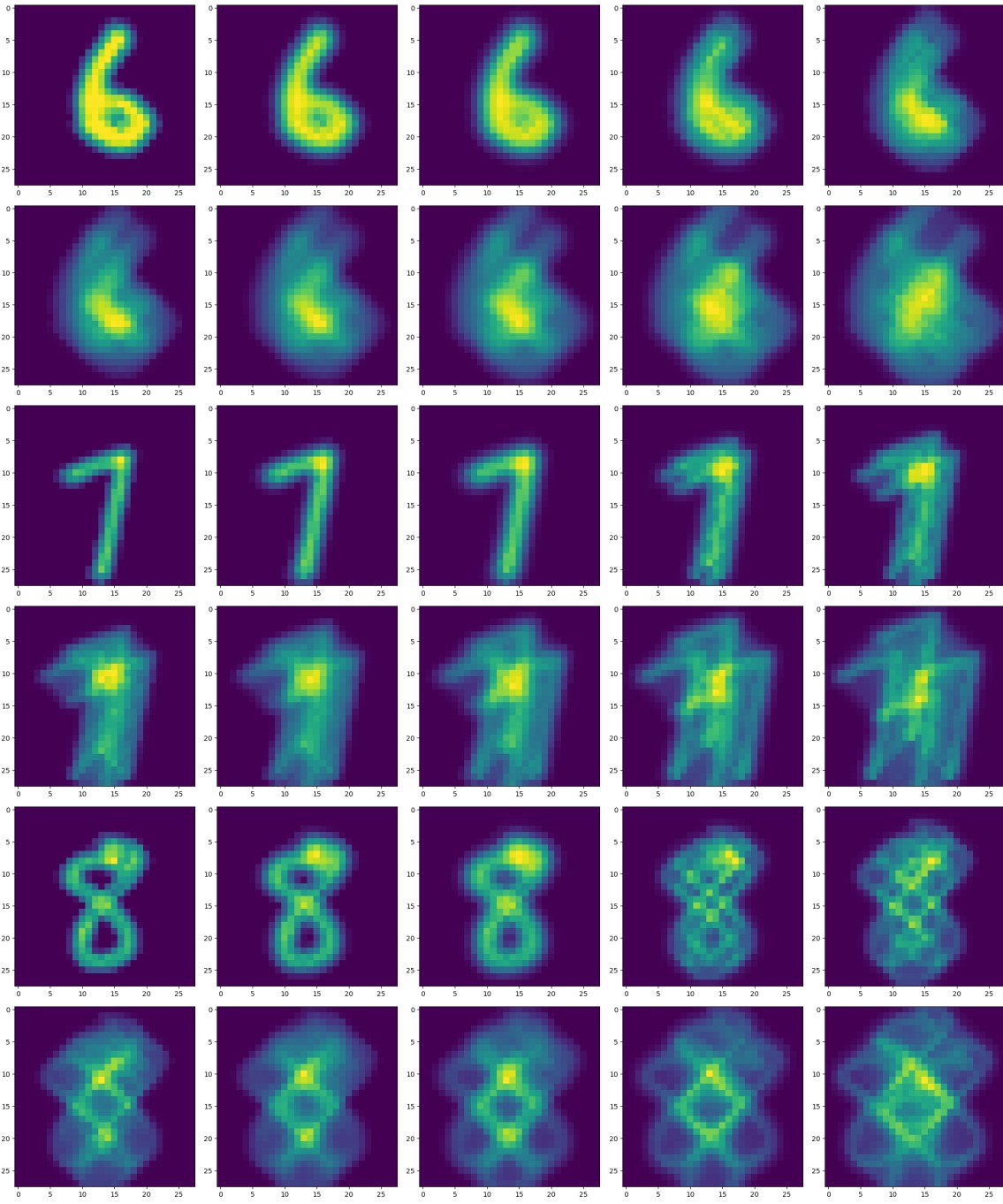

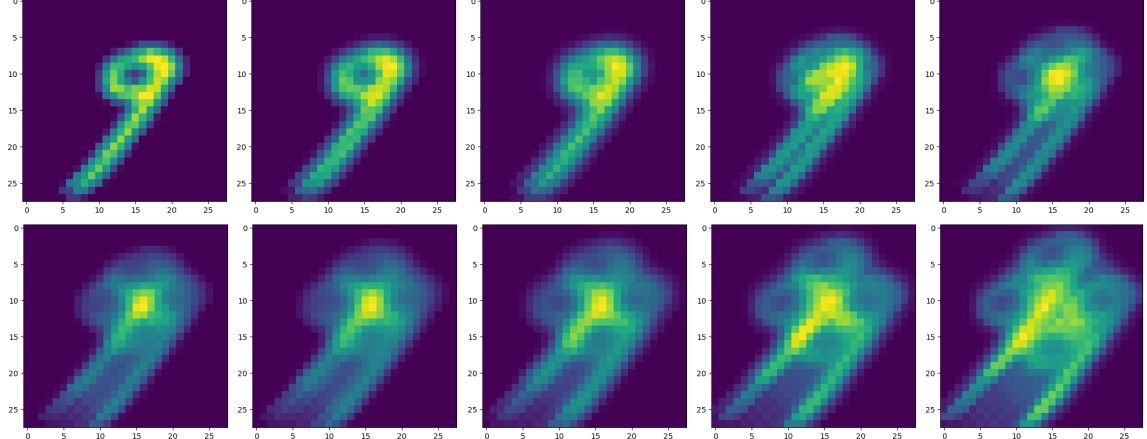

**Classical diffusion on MNIST dataset (equal to low pass filter as discussed in the paper):**

(only step 10)

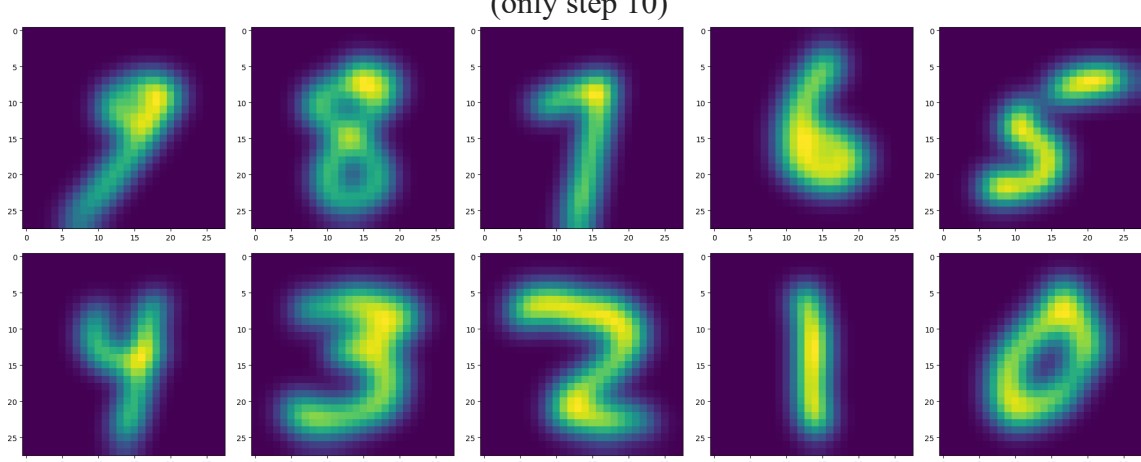