# OpenReview forum: "Beyond Classical Diffusion: Ballistic Graph Neural Network"
_ICLR.cc/2020/Conference — Reject_

### Official Review · AnonReviewer3 · 2019-10-21
**Official Blind Review #3**

**Rating:** 3

**Review:**

This paper proposed a new diffusion operation for the graph neural network. Specifically, the ballistic graph neural network does not require to calculate any eigenvalue and can propagate exponentially faster comparing to traditional graph neural network. Extensive experiments have been conducted to verify the performance of the proposed method.

1. The motivation of this method is to accelerate the diffusion speed in a graph. However, as we know, a very severe issue of graph neural network is the over-smoothness issue. The reason is that, in the high layer, the node feature is diffused to far neighbours.  When using the proposed ballistic filter, node features diffuse much faster than the regular GNN. Thus, the over-smoothness will appear in the shallow layer very fast. As a result, we cannot use many layers so that the non-linearity of deep neural networks cannot be fully utilized. Thus, is it necessary to accelerate the diffusion speed for graph neural network?

2. There is only one dataset for  the comparison of the performance of different graph neural networks. More datasets are needed to thoroughly verify the performance of the proposed ballistic graph neural network.

3. Is it possible to slow down the diffusion speed with the proposed ballistic filter?




**Experience Assessment:**

I have published in this field for several years.

**Review Assessment: Checking Correctness Of Derivations And Theory:**

I assessed the sensibility of the derivations and theory.

**Review Assessment: Checking Correctness Of Experiments:**

I assessed the sensibility of the experiments.

**Review Assessment: Thoroughness In Paper Reading:**

I read the paper thoroughly.

---

### Official Review · AnonReviewer1 · 2019-10-23
**Official Blind Review #1**

**Rating:** 3

**Review:**

The paper "Beyond Classical Diffusion: Ballistic Graph Neural Network" tackles the problem of graph vertices representation. While most existing works rely on classical random walks on the graph, the paper proposes to cope with the "speed of diffusion" problem by introducing ballistic walk.

I noticed the comment of the authors that gives a correction for the introduction. But even with it the paper remains very cryptic, with very few pointers to help the reader in understanding the contribution. The introduction (even corrected) is very abrupt and it is very difficult to understand the problem that the authors propose to attack. The problem is that authors start with mathematical discussions before presenting the manipulated concepts and formalizing the adressed problem. I only understood the adressed problem after seing which are the baselines the proposal is compared with in section 4.2. Also, the introduction does not introduce the proposal at all.

A symptomatic example of the lack of paper positioning is the Related Works section which does not even give a single reference !  A related work section with no related works in it appears to have a limited interest to me...  This section should at least introduce other works in the field of graph embedding, such as those reported as baselines. It would also greatly help to understand the contribution of the paper. Also, the ballistic concept is not introduced at all in section 4. Where does this term comes from ? The proposed approach is completely cryptic, with clearly not enough definition of the notations the algorithm deals with. A global view of the approach, from the input graph to the final representation, would also be required to help the reader to understand the proposal. If the contribution is only a new kind of random walk on a graph, is ICLR the good targeted venue ? If authors think so, they should present their contribution in a representation learning perspective, which would highlight the importance of this new walk for the graph representation learning process.


From my point of view, without a full re-writting of the paper, this work cannot be published in a conference like ICLR.

**Experience Assessment:**

I have published one or two papers in this area.

**Review Assessment: Checking Correctness Of Derivations And Theory:**

I did not assess the derivations or theory.

**Review Assessment: Checking Correctness Of Experiments:**

I assessed the sensibility of the experiments.

**Review Assessment: Thoroughness In Paper Reading:**

I made a quick assessment of this paper.

---

### Official Review · AnonReviewer2 · 2019-10-27
**Official Blind Review #2**

**Rating:** 1

**Review:**

This paper was extremely hard to read or comprehend. It’s riddled with typos, inaccurate notations and undefined variables (see below for a sampling). The authors will need to significantly polish and improve the presentation of the paper.

After a few forward and backward passes through the paper, I was able to gather the following high level ideas about the paper:
(1) This paper is somewhat related to the Defferard et. al, 2016 in that the authors want to define a propagation filter for graph neural networks.
2) This proposed filter known as “ballistic filter” should have the property of allowing fast diffusion through the network.
(3) The authors claim that the ballistic kernel diffuses @ O(k) as compared to O(\sqrt k) when compared to traditional GCNs, where k is the number of propagation steps.
(4) The authors additionally claim that their approach needs one-third the number of parameters.
(5) The authors provide some plots to visualize the linear diffusion rate of their proposed filter.

--- Issues and clarifications ---
- Sec 3, Eq 1 seems to have been taken from Eq 1 in Defferard et. al, however there’s no reference to it and the terms g, U, etc. are not defined.
 - Sec 4, Algo 1 contains the main core of the proposed algorithm, but it’s only defined for the 2D grid case. The notation therein is extremely unclear. What is H_space, H_c? How does one sample \hat{O}_coin.? The net result is that algorithm is undefined. Without a clear definition of the algorithm, it’s completely unclear what the proposed method does.
 - Sec 4.2 is completely unparseable. What is problem setting? What is the metric? How have the baselines been implemented? How has data been split for training/testing?
 - Section 5 mentions that one-third params are used to get 97% but no details are provided as to how less params are consumed.
 - How is figure 7 generated?
 - Sec 8, feel totally unrelated to the paper. There are a whole bunch of random, unmotivated diffusion equations  Eq 6, mentions “.. \hat{g}(f) decreases as f increases and thus can be seen as a low pass filter…” . This is not true from the formula.


 --- A sampling of typos ---
Sec 4.1, .. consisits …
Sec 5 “REVISIT” -> “REVISITING”
Figure 6, text, “cassical”
Sec 6.2 title, “SUMMAY”
Sec 8  “aggreated”
Sec 8  t=\-tau to -\tau
Several typos with Hardmard, Hadmard instead of Hadamard.

Overall, the major criticisms of this paper:
 - The proposed algorithm is not clear.
 - The authors need much more experimentation to bolster their claims in the paper. It’s completely unclear if fast diffusion even if it were possible will help GNNs perform better on a diverse set of tasks.
 - The paper needs a lot more polish and proof reading to make this paper presentable.


**Experience Assessment:**

I have published one or two papers in this area.

**Review Assessment: Checking Correctness Of Derivations And Theory:**

I assessed the sensibility of the derivations and theory.

**Review Assessment: Checking Correctness Of Experiments:**

I assessed the sensibility of the experiments.

**Review Assessment: Thoroughness In Paper Reading:**

I read the paper at least twice and used my best judgement in assessing the paper.

---

### Decision · Program_Chairs · 2019-12-19

**Decision:**

Reject

**Comment:**

This submission has been assessed by three reviewers who scored it 3/1/3, and they have remained unconvinced after the rebuttal. The main issues voiced are the difficult readability of the paper, cryptic at times due to a mix of physical and DL notations, and a lack of sufficient experimentation to support all claims. The reviewers acknowledge the authors' efforts to resolve the main issues but find these efforts insufficient. Thus, this paper cannot be accepted to ICLR2020.